# Integrative Analysis of Transcriptomic and Lipidomic Profiles Reveals a Differential Subcutaneous Adipose Tissue Mechanism among Ningxiang Pig and Berkshires, and Their Offspring

**DOI:** 10.3390/ani13213321

**Published:** 2023-10-25

**Authors:** Xiaoxiao Deng, Yuebo Zhang, Gang Song, Yawei Fu, Yue Chen, Hu Gao, Qian Wang, Zhao Jin, Yulong Yin, Kang Xu

**Affiliations:** 1Laboratory of Animal Nutrition Physiology and Metabolism, The Institute of Subtropical Agriculture, The Chinese Academy of Sciences, Changsha 410125, China; dengxiaoxiao@isa.ac.cn (X.D.); fuyw2020@163.com (Y.F.); 18837025618@126.com (Y.C.); 2Guangdong Laboratory for Lingnan Modern Agriculture, Guangzhou 510642, China; zambofd@gmail.com (Y.Z.); sg19971109@163.com (G.S.); gaohu_20190008@163.com (H.G.); wangq0130@163.com (Q.W.); jz2725552543@163.com (Z.J.); 3CAS Key Laboratory of Agro-Ecological Processes in Subtropical Region, Institute of Subtropical Agriculture, Changsha 410125, China; 4Hunan Provincial Key Laboratory for Genetic Improvement of Domestic Animal, College of Animal Science and Technology, Hunan Agricultural University, Changsha 410125, China

**Keywords:** pigs, subcutaneous adipose tissue, transcriptomic, lipidomic, KEGG pathways

## Abstract

**Simple Summary:**

Understanding differential mechanisms of the subcutaneous adipose tissue between indigenous breeds and modern pig breeds is important. After transcriptome sequencing and liposome detection of the subcutaneous adipose tissue in Ningxiang pigs, Berkshires and F_1_ offspring, we retrieved differentially expressed genes (DEGs) and significantly changed lipids (SCLs) in different groups. Integrative analysis of transcriptomic and lipidomic profiles and KEGG annotation revealed that the difference in fat deposition among Ningxiang pig, Berkshires, and F_1_ offspring may be caused by differences in the expression patterns of key genes in multiple enriched KEGG pathways. This study identified the key genes and lipids and also provided insights into selection for backfat thickness and the fat composition of adipose tissue for pig breeding.

**Abstract:**

Adipose tissue composition contributes greatly to the quality and nutritional value of meat. Transcriptomic and lipidomic techniques were used to investigate the molecular mechanisms of the differences in fat deposition in Ningxiang pigs, Berkshires and F_1_ offspring. Transcriptomic analysis identified 680, 592, and 380 DEGs in comparisons of Ningxiang pigs vs. Berkshires, Berkshires vs. F_1_ offspring, and Ningxiang pigs vs. F_1_ offspring. The lipidomic analysis screened 423, 252, and 50 SCLs in comparisons of Ningxiang pigs vs. Berkshires, Berkshires vs. F_1_ offspring, and Ningxiang pigs vs. F_1_ offspring. Lycine, serine, and the threonine metabolism pathway, fatty acid biosynthesis and metabolism-related pathways were significantly enriched in comparisons of Berkshires vs. Ningxiang pigs and Berkshires vs. F_1_ offspring. The DEGs (*PHGDH*, *LOC110256000*) and the SCLs (phosphatidylserines) may have a great impact on the glycine, serine, and the threonine metabolism pathway. Moreover, the DEGs (*FASN*, *ACACA*, *CBR4*, *SCD*, *ELOV6*, *HACD2*, *CYP3A46*, *CYP2B22*, *GPX1*, and *GPX3*) and the SCLs (palmitoleic acid, linoleic acid, arachidonic acid, and icosadienoic acid) play important roles in the fatty acid biosynthesis and metabolism of fatty acids. Thus, the difference in fat deposition among Ningxiang pig, Berkshires, and F_1_ offspring may be caused by differences in the expression patterns of key genes in multiple enriched KEGG pathways. This research revealed multiple lipids that are potentially available biological indicators and screened key genes that are potential targets for molecular design breeding. The research also explored the molecular mechanisms of the difference in fat deposition among Ningxiang pig, Berkshires, and F_1_ pigs, and provided an insight into selection for backfat thickness and the fat composition of adipose tissue for future breeding strategies.

## 1. Introduction

Pork is a major source of amino acids and fatty acids, especially in China. Healthy, safe, and nutritious high-quality and specially flavored pork is more and more popular among consumers, and in huge demand. Adipose tissue is a dynamic metabolic and endocrine organ, which also stores fat and secretes biologically active substances, such as leptin and lipid mediators [1]. The accumulation of body fat may be the result of the balance among the absorption of fat, de novo fat synthesis, and fat metabolism and catabolism [2]. The fatty acid composition of adipose tissue contributes greatly to the flavor, quality, and nutritional value of meat [3]. The lipid content and the fatty acid composition in adipose tissue are strongly affected by the genotype and nutrient supply [4].

Chinese indigenous breeds, which have excellent meat quality and a special flavor, are more and more popular with consumers and researchers compared with modern pig breeds [5,6]. However, their low growth rate and a low proportion of lean meat limit the large-scale breeding of indigenous breeds. Fat deposition exhibits huge differences between indigenous breeds and modern pig breeds [7]. Selective breeding for a high proportion of lean meat and low backfat thickness has resulted in a reduced lipogenic potential in the modern pig breeds, while indigenous breeds have preserved a high fat deposition capacity [4]. The cultivated breeds with an indigenous pig lineage, which have a high growth rate, a high proportion of lean meat, low backfat thickness, and excellent flavor are required to meet market demand. The cultivated breeds are cultivated by hybrid breeding between indigenous breeds and modern pig breeds, such as Xiangcun Black pig (cultivated by hybrid breeding between Taoyuan Black pig and Duroc) and Sutai pig (cultivated by hybrid breeding between Taihu pig and Duroc). Nowadays, the question of how to apply multi-omics data to accelerate pig breeding is an important problem. Recently, multi-omics association analysis brought about opportunities for exploring the molecular mechanisms of complex traits and for more efficient mining of candidate genes for complex traits [8,9,10,11]. Clarifying the molecular mechanisms of certain traits and candidate gene resources may contribute to pig breeding [12]. The molecular mechanisms of the differences in adipose tissue between indigenous breeds and modern pig breeds is complex. Several studies compared the lipogenic enzyme activities of subcutaneous adipose tissue in multiple groups (Basque vs. Large White pigs, Alentejano vs. Large White pigs, Meishan vs. Large White pigs), and the results showed that the activity and gene expression of acetyl-CoA carboxylase 1 (*ACACA*) and glucose-6-phosphate dehydrogenase (*G6PDH*) are greater in Basque and Alentejano than in Large White, whereas the activity and gene expression of stearoyl-coA desaturase (*SCD*) is lower in Meishan pig than in Large White [13,14,15]. Thus, differences in the expression and activity of these key genes among different varieties may contribute to the phenotypic differences.

The Ningxiang pig, one of four well-known Chinese indigenous breeds, has abundant polyunsaturated fatty acid content and superior meat quality [16]. The Ningxiang pig is a typical fatty breed, with a slow growth rate and a strong fat deposition ability. The Berkshires is a typical lean pig breed, famous for its high proportion of lean meat and fast growth rate. When Ningxiang pigs acted as the male parent in a cross with Berkshires, the meat quality of the F_1_ pigs was superior. Compared with Ningxiang pigs, the F_1_ offspring had less backfat thickness. Obvious phenotypic differences existed in the subcutaneous adipose tissue among Ningxiang pig, Berkshires, and F_1_ offspring. Multi-omics association analysis has proven to be an effective method for investigating the molecular mechanisms of fat deposition [17]. However, there are scant multi-omics data on the differences among Ningxiang pig, Berkshires, and their offspring, and the molecular mechanisms of the difference in fat deposition among Ningxiang pig, Berkshires, and their offspring are still unknown. Therefore, this study aimed to research the molecular mechanisms of the difference in fat deposition among Ningxiang pig, Berkshires, and F_1_ offspring, and to identify the key genes and lipids. This study also provided insights into selection for backfat thickness and the fat composition of adipose tissue for pig breeding. 

## 2. Materials and Methods

### 2.1. Ethics Statement

The animal experiments were conducted according to the animal welfare requirements and approved by the Animal Protocol Review Committee of the Institute of Subtropical Agriculture, Chinese Academy of Science (No. ISA-2023-0020).

### 2.2. Animals

In total, 42 pigs (14 purebred Ningxiang pigs, 14 Berkshire pigs, and 14 hybrids of Ningxiang pig × Berkshires) from the same farm were used in this study. All pigs were raised on the same farm and housed under standard management conditions by the Dalong Livestock Co. Ltd., in Changsha China, with the same diet and free access to water. All pigs were healthy. The subcutaneous adipose tissue samples from these pigs were collected from Chu Weixiang Slaughtering and Cutting Plant in Ningxiang City, Hunan Province. We randomly selected six Ningxiang pigs with a left carcass weight of 32–38 kg, six Berkshires with a left carcass weight of 41–47 kg, and six hybrid progenies of Ningxiang pig × Berkshires (F_1_ offspring) with a left carcass weight of 37–41 kg, and collected the subcutaneous adipose from between the 6th and 11th rib samples from each pig. The samples were crushed and packed, and then stored in a freezer at −80 °C with a constant room temperature and humidity. 

### 2.3. RNA Extraction

We selected 12 samples of subcutaneous adipose from four Ningxiang pigs, four Berkshires, and four F_1_ offspring, respectively. These 12 samples of subcutaneous adipose were used to detect the transcriptomic profiles. The total RNA was extracted using the TRIzol reagent (Life Technologies, Waltham, MA, USA). The concentration and purity of the RNA was measured using NanoDrop 2000 (Thermo Fisher Scientific, Wilmington, DE, USA). RNA integrity was assessed using the RNA Nano 6000 Assay Kit of the Agilent Bioanalyzer 2100 system (Agilent Technologies, Santa Clara, CA, USA). A total of 1 μg RNA per sample was used as input material for the preparation of RNA samples.

### 2.4. mRNA Sequencing and Transcriptomic Data Analysis

RNA-seq was performed by Biomarker Technology Co., Ltd. (Beijing, China). Sequencing libraries were generated using the NEBNext UltraTM RNA Library Prep Kit for Illumina (NEB, San Diego, CA, USA), and then the library’s quality was assessed on the Agilent Bioanalyzer 2100 system. After cluster generation, the library preparations were sequenced on an Illumina platform, and paired-end reads were generated. The raw data were filtered using the FastQC program (http://www.bioinformatics.babraham.ac.uk/projects/fastqc/, accessed on 8 September 2022). Clean reads were obtained by removing reads containing adapters, reads containing poly-N, and low-quality reads from the raw data. At the same time, the Q20, Q30, GC-content, and sequence duplication level of the clean data were calculated to ensure the quality of the clean reads was high. These clean reads were then mapped to the pig reference genome Sus scrofa11.1 (http://www.ensembl.org/info/data/ftp/index.html, accessed on 10 September 2022), by Hisat2 software (http://www.ccb.jhu.edu/software/hisat, accessed on 10 September 2022). Only reads with a perfect match or one mismatch were analyzed further and annotated on the basis of the pig reference genome. Differential expression analysis in the subcutaneous adipose tissue group (12 samples of subcutaneous adipose from four Ningxiang pigs, four Berkshires, and four F_1_ offspring) was performed using the DESeq2 [18]. The differential gene screening conditions were: |log2foldchange| > 1, and false discovery rate (FDR) < 0.05. The gene ontology (GO) enrichment analysis of the DEGs was implemented by the GOseq R package [19]. Version 3.0 of KOBAS [20] software was used to test the statistical enrichment of differentially expressed genes in the KEGG pathways. *p*-values were adjusted by the FDR method, and adjusted *p*-values < 0.05 were considered significant. RT-qPCR was performed to validate the reliability of RNA-seq. Specific primers of 11 genes were designed based on the gene sequences of pigs (Appendix A).

### 2.5. Lipids Extraction and LC-MS/MS Analysis

We selected 18 samples of subcutaneous adipose from six Ningxiang pigs, six Berkshires, and six F_1_ offspring, respectively. These 18 samples of subcutaneous adipose were used to detect the lipidomic profiles. A 100 μL sample was mixed with 1 mL of the extraction solvent (MTBE: MeOH = 3:1, *v*/*v*) containing an internal standard mixture, and then vortexed for 15 min and sonicated in an ice bath for 10 min. Then, 200 μL of ultra-pure water were added. Next, 200 μL of the extraction solution were collected and evaporated using a vacuum concentrator. The dry extract was reconstituted using 400 μL Mobile Phase B prior to LC-MS/MS analysis. The sample extracts were analyzed using an LC-ESI-MS/MS system (UPLC, ExionLC AD, https://sciex.com.cn/, accessed on 10 September 2022; MS, QTRAP^®^ System, https://sciex.com/, accessed on 10 September 2022). The analytical conditions were as follows, UPLC: column, Thermo Accucore™ C30 (2.6 μm, 2.1 mm × 100 mm i.d.); Solvent System, A: acetonitrile/water (60/40, *v*/*v*, 0.1% formic acid, 10 mmol/L ammonium formate), Solvent System B: acetonitrile/isopropanol (10/90 *v*/*v*, 0.1% formic acid, 10 mmol/L ammonium formate); gradient program, A/B (80:20, *v*/*v*) at 0 min, 70:30 *v*/*v* at 2 min, 40:60 *v*/*v* at 4 min, 15:85 *v*/*v* at 9 min, 10:90 *v*/*v* at 14 min, 5:95 *v*/*v* at 15.5 min, 5:95 *v*/*v* at 17.3 min, 80:20 *v*/*v* at 17.3 min, 80:20 *v*/*v* at 20 min; flow rate, 0.35 mL/min; temperature, 45 °C; injection volume: 2 μL. The effluent was alternatively connected to an ESI-triple quadrupole-linear ion trap (QTRAP)-MS. LIT and triple quadrupole (QQQ) scans were acquired on a triple quadrupole-linear ion trap mass spectrometer (QTRAP). The ESI source operation parameters were as follows: ion source, turbo spray; source temperature, 500 °C; ion spray voltage (IS), 5500 V (Positive; −4500 V (negative)); ion source gas 1 (GS1), gas 2 (GS2), and curtain gas (CUR) were set at 45, 55, and 35 psi, respectively; the collision gas (CAD) was medium. Instrument tuning and mass calibration were performed with 10 and 100 μmol/L of polypropylene glycol solutions in QQQ and LIT modes, respectively. The QQQ scans were acquired as MRM experiments with the collision gas (nitrogen) set to five psi. The DP and CE for individual MRM transitions were carried out with further optimization of DP and CE. A specific set of MRM transitions was monitored for each period according to the lipids eluted within this period.

### 2.6. Statistical Analysis

Unsupervised PCA (principal component analysis) was performed by the statistical function prcomp of R (version 3.5.1, www.r-project.org, accessed on 10 September 2022). The data were unit-variance scaled before the unsupervised PCA. The variable importance in the projection (VIP) was calculated in the OPLS-DA model, and generated using the R package MetaboAnalystR (Version 1.01). Differential lipids were determined by VIP (VIP ≥ 1) and absolute log2FC (|Log2FC| ≥ 1.0). In order to avoid overfitting, a permutation test (200 permutations) was performed. Then, the differential lipids were annotated using the KEGG Compound database (http://www.kegg.jp/kegg/compound/, accessed on 10 September 2022), and mapped into biochemical pathways through KEGG enrichment analysis. Pathways with significantly regulated metabolites mapped to them were then fed into MSEA (metabolite sets enrichment analysis), and their significance was determined by the hypergeometric test’s *p*-values. Heat maps of DEGs were generated using the Metware Cloud, a free online platform for data analysis (https://cloud.metware.cn, accessed on 10 September 2022).

## 3. Results

### 3.1. The Backfat Thickness of Ningxiang Pigs, Berkshires and F_1_ Offspring

We measured the backfat thickness of pigs of the same age and found that the backfat thickness of Ningxiang pigs was significantly higher than that of Berkshires and F_1_ offspring, and the backfat thickness of F_1_ offspring was significantly higher than that of Berkshires (Figure 1).

### 3.2. Lipidomic Data Analysis

To investigate the differences in the lipid metabolite compositions of the subcutaneous adipose tissue among Ningxiang pig, Berkshires, and F_1_ offspring, LC-MS/MS analysis was performed. In total, 1293 lipids were detected, including 506 glycerphospholipids (GP), 421 glycerolipids (GL), 226 sphingolipids (SP), 88 triglycerides (TG), 40 fatty acyls (FA), 10 sterol lipids (ST), and two prenol lipids (PR). In the detected lipid subclasses, triglycerides (TG), diglycerides (DG), phosphatidylethanolamine (PE), phosphatidylcholine (PC), phosphatidylserine (PS), and ceramide (Cer-AS) were abundant in the subcutaneous adipose tissue (Appendix A). Principal component analysis (PCA) indicated that the three different groups were partly separated in the plots of PC1 × PC2 scores (Appendix A). Further orthogonal partial least squares discriminant analysis (OPLS-DA) showed that there were apparent distinctions among the three groups, and small distinctions within the groups (Appendix A). Moreover, the cluster heat map of the overall sample, which was derived from the Z-score for the expression levels of all samples, and K-means analysis suggested that the relative lipid content of the subcutaneous adipose tissue of Berkshires was higher than that of Ningxiang pigs and F_1_ offspring for the lipids detected in this study (Appendix A). Next, the SCLs were filtered and determined by their variable importance in projection (VIP) value ≥ 1.0 and |Log2(fold change)| ≥ 1.0. In total, 423 SCLs were screened in the subcutaneous adipose tissue for the group of Berkshires vs. Ningxiang pigs. Compared with Berkshires, Ningxiang pigs had 412 downregulated lipids, primarily GPs, GLs, SPs, DGs, FAs, STs, and PR (238, 93, 44, 24, 10, two, and one, respectively), but only 11 upregulated lipids, including eight TGs, two SPs and one GP, in the subcutaneous adipose tissue (Figure 2A). Moreover, 252 SCLs were markedly changed in the subcutaneous adipose tissue of F_1_ offspring compared with that of Berkshires. The F_1_ offspring had 251 downregulated lipids, containing three FAs, 13 GLs, 219 GPs, 12 SPs and four TGs, but only one SP was upregulated in the subcutaneous adipose tissue (Figure 2B). For the comparison between Ningxiang pigs and F_1_ offspring, 50 SCLs were screened. Compared with Ningxiang pigs, F_1_ offspring had just seven downregulated lipids including four GPs, one SP, one TG, and one GL, and 43 upregulated lipids containing 12 GLs, 11 GPs, 11 SPs, 4 STs, 3 FAs, and 2 TGs in the subcutaneous adipose tissue (Figure 2C).

Subsequently, KEGG enrichment analysis was performed to uncover the biological mechanisms of the differences in subcutaneous adipose tissue among Ningxiang pigs, Berkshires, and F_1_ offspring. With respect to the comparison between Berkshires and Ningxiang pigs, 423 SCLs were significantly annotated into 108 pathways, and the majority of SCLs were involved in fatty acid and amino acid metabolism-related pathways such as glycerophospholipid metabolism, linoleic acid metabolism, arachidonic acid metabolism, alpha-linolenic acid metabolism, inositol phosphate metabolism, and glycine, serine, and threonine metabolism (Figure 2D). For the comparison between Berkshires and F_1_ offspring, 252 SCLs were significantly annotated into 38 pathways, including glycerophospholipid metabolism, linoleic acid metabolism, arachidonic acid metabolism, alpha-linolenic acid metabolism, glycine, serine and threonine metabolism, and the cAMP signaling pathway (Figure 2E). For the comparison between Ningxiang pigs and F_1_ offspring, and 50 SCLs were significantly annotated into 31 pathways, including teichoic acid biosynthesis, inositol phosphate metabolism, steroid biosynthesis, ovarian steroidogenesis, glycerolipid metabolism, and the phosphatidylinositol signaling system (Figure 2F). Thus, there were 423, 252, and 50 SCLs up- or downregulated between Berkshires and Ningxiang pigs, between Berkshires and F_1_ offspring, and between Ningxiang pigs and F_1_ offspring, respectively (Figure 3A). In total, 13 SCLs were commonly identified in the three pairwise comparisons (Figure 3A). Additionally, there were 12, nine, and six KEGG pathways (*p* < 0.05) enriched in the comparison between Berkshires and Ningxiang pigs, between Berkshires and F_1_ offspring, and between Ningxiang pigs and F_1_ offspring, respectively (Figure 3B). Interestingly, nine pathways were commonly enriched in Berkshires vs. Ningxiang pigs and in Berkshires vs. F_1_ offspring, including seven pathways (KO05231, KO00563, KO00591, KO00592, KO00590, and KO00564) involved in lipid metabolism; one pathway (KO00260) involved in glycine, serine, and threonine metabolism; and one pathway (KO04723) involved in retrograde endocannabinoid signaling (Figure 3B). Moreover, three pathways were commonly enriched in Berkshires vs. Ningxiang pigs and in Ningxiang pigs vs. F_1_ offspring, including inositol phosphate metabolism (kO00562), teichoic acid biosynthesis (KO00552), and the phosphatidylinositol signaling system (KO04070) (Figure 3B). Moreover, three pathways were uniquely enriched in Ningxiang pigs vs. F_1_ offspring, including steroid biosynthesis (KO00100), ovarian steroidogenesis (KO04913), and bile secretion (KO04976).

### 3.3. Transcriptome Data Analysis

Transcriptome sequencing of the subcutaneous adipose tissue of Ningxiang pigs, Berkshires, and F_1_ offspring was performed, in order to verify the reliability of the results, and their correlations were analyzed. The correlation analysis indicated a strong correlation within groups, and that the correlations between the groups were slightly lower (Appendix A). As shown in the heatmap (Appendix A), the expression patterns of DEGs were consistent with the sample groups. We counted the DEGs among the three different groups. We found 395 upregulated genes and 285 downregulated genes, and a total of 680 DEGs in the subcutaneous adipose tissue of the group of Berkshires vs. Ningxiang pigs (Figure 4A). There were 317 upregulated genes, and 275 downregulated genes, and a total of 592 DEGs in the subcutaneous adipose tissue of F_1_ offspring compared to that of Ningxiang pigs (Figure 4B). There were 230 upregulated genes, and 154 downregulated genes, and a total of 384 DEGs in the subcutaneous adipose tissue of F_1_ offspring compared to that of Berkshires (Figure 4C). Interestingly, the number of DEGs from the comparison between Berkshires and Ningxiang pigs was the greatest among the three different comparisons.

On the basis of the identified DEGs, functional annotation and classification were performed via GO analysis. DEGs were classified into three primary categories, namely biological process (BP), cellular components (CC), and molecular functions (MF). For Berkshires vs. Ningxiang pigs, GO enrichment analysis of DEGs showed that the enriched biological processes were mainly enriched in the process related to xenobiotic metabolism, organic acid metabolism and L-serine biosynthesis (Appendix A); the enriched cellular components were mainly enriched in collagen, lipid droplets and cytoplasm (Appendix A); and the enriched molecular functions were mainly enriched in lipid binding and transmembrane receptor protein serine/threonine kinase activity (Appendix A). For Ningxiang pigs vs. F_1_ offspring, GO enrichment analysis of DEGs, the enriched biological processes were mainly processes related to translation, brown fat cell differentiation and ATP synthesis coupled proton transport (Appendix A); the enriched cellular components were mainly mitochondrial respiratory chain Complex I, collagen trimer and extracellular matrix (Appendix A); and the enriched molecular functions were mainly for cytochrome-c oxidase activity, structural constituents of the ribosome and NADH dehydrogenase activity (Appendix A). For the groups of Berkshires vs. F_1_ offspring, GO enrichment analysis of the DEGs, the enriched biological processes were mainly processes related to the response to xenobiotic stimulus, immune response and the specification of stem cell fate (Appendix A); the enriched cellular components were mainly the extracellular region, cytoplasm, and collagen trimers (Appendix A); the enriched molecular functions were mainly in chemokine activity, phosphoglycerate dehydrogenase activity, and sterol esterase activity (Appendix A). Then, the DEGs were annotated with the KEGG pathways. There were 35, 22, and 19 KEGG pathways (*p* < 0.05) enriched in the comparison between Berkshires and Ningxiang pigs, Berkshires and F_1_ offspring, and Ningxiang pigs and F_1_ offspring, respectively (Figure 5). Interestingly, eight pathways were commonly enriched in the comparison between Berkshires and Ningxiang pigs, and between Berkshires and F_1_ offspring, six pathways were commonly enriched in comparison between F_1_ offspring and Ningxiang pigs, and between Berkshires and F_1_ offspring; and 11 pathways were commonly enriched in comparison between Berkshires and Ningxiang pigs and between Ningxiang pigs and F_1_ offspring (Figure 5). For Berkshires vs. Ningxiang pigs, KEGG analysis of the subcutaneous adipose tissue yielded significant pathways, such as the carbon metabolism, fatty acid metabolism, and fatty acid biosynthesis pathways (Figure 6A, Appendix A). For Ningxiang pigs vs. F_1_ offspring, KEGG analysis revealed that the significant pathways were closely related to oxidative phosphorylation, thermogenesis and protein digestion and absorption (Figure 6B). For the groups of Berkshires vs. F_1_ offspring, KEGG analysis indicated that significant pathways were closely related to the TNF signaling pathway, cysteine and methionine metabolism and glycine, serine, and threonine metabolism (Figure 6C). Additionally, because of limitations of the small sample sizes (12 samples of subcutaneous adipose from four Ningxiang pigs, four Berkshires and four F_1_ offspring were used to detect the transcriptomic profiles, samples of four animals from each genetic grouping may not be sufficient for the analysis performed), the genes identified as DEGs could have occurred by chance, and, therefore, the results of the transcriptome data analysis are speculative.

### 3.4. Joint Analysis of the Transcriptome and Lipidome Data

A joint analysis of the transcriptome and lipidome data were carried out to identify the origins of the transcriptomic and lipidomic differences among Ningxiang pigs, Berkshires, and F_1_ offspring. In total, 680 DEGs (395 upregulated and 285 downregulated) and 423 SCLs (11 upregulated and 412 downregulated) were identified in Berkshires vs. Ningxiang pigs, 384 DEGs (230 upregulated and 154 downregulated) and 252 SCLs (1 upregulated and 251 downregulated) were identified in the group of Berkshires vs. F_1_ offspring and 592 DEGs (317 upregulated and 275 downregulated) and 50 SCLs (43 upregulated and 7 downregulated) were identified in Ningxiang pigs vs. F_1_ offspring. Furthermore, pathway annotation was performed and the significant pathways were annotated with the DEGs and SCLs for Berkshires vs. Ningxiang pigs (Figure 7A), Berkshires vs. F_1_ offspring (Figure 7B), and Ningxiang pigs vs. F_1_ offspring (Figure 7C), respectively. Interestingly, the most significantly enriched pathways for the DEGs and SCLs in both the comparison between Berkshires and Ningxiang pigs, and between Berkshires and F_1_ offspring, were the glycine, serine, and threonine metabolism pathways (Figure 7A,B).

### 3.5. Screening of the Candidate Genes and Lipids with the Combined Transcriptomic and Lipidomic Data

KO00260, the lycine, serine, and threonine metabolism pathways, significantly enriched in Berkshires vs. Ningxiang pigs and Berkshires vs. F_1_ offspring, respectively. For Berkshires vs. Ningxiang pigs, DEGs related to KO00260, the lycine, serine, and threonine metabolism pathways, such as phosphoglycerate dehydrogenase (*PHGDH*), phosphoglycerate dehydrogenase like protein (*LOC100156167*), and glycine hydroxymethyltransferase (*SHMT1*) were significantly downregulated, whereas phosphoserine phosphatase (*PSPH*) and membrane primary amine oxidase-like (*LOC110256000*) were significantly upregulated (Figure 8A). In addition, the content of the SCLs related to KO00260, (32 phosphatidylserines) of the subcutaneous adipose tissue were reduced in Berkshires vs. Ningxiang pigs (Appendix A). For the group of Berkshires vs. F_1_ offspring, *PHGDH* was significantly downregulated, whereas, *LOC110256000* and serine dehydratase (*SDS*) were significantly upregulated (Figure 8B). Moreover, the content of the SCLs related to KO00260, (31 phosphatidylserines) of the subcutaneous adipose tissue were reduced in Berkshires vs. F_1_ offspring (Appendix A). *PHGDH* is a key enzyme of the serine synthesis pathway [21]. In this study, *PHGDH* was significantly downregulated in the subcutaneous adipose tissue of Ningxiang pigs and F_1_ offspring compared to that of Berkshires, and the content of phosphatidylserines in the subcutaneous adipose tissue of Ningxiang pigs and F_1_ offspring was reduced compared to that of Berkshires. The SCLs and DEGs (Pearson correlation coefficient > 0.8 and *p*-value < 0.05) in the KEGG pathways were selected to study the correlation between the SCLs and DEGs. The results indicated that the expression of *PHGDH* and *LOC100156167* were positively correlated with the content of phosphatidylserines, while the expression of *LOC110256000* was negatively correlated with the content of phosphatidylserines in the group of Berkshires vs. Ningxiang pigs (Figure 8C) and in Berkshires vs. F_1_ offspring (Figure 8D).

For Berkshires vs. Ningxiang pigs, DEGs related to the KO00061 (fatty acid biosynthesis) pathways, such as, fatty acid synthase (*FASN*), acetyl-CoA carboxylase (*ACACA*), carboxyl reductase 4 (*CBR4*), and medium chain acyl hydrolase (*MCH*) were upregulated (Figure 9A). Moreover, the content of the SCLs related to the KO00061 pathways of the subcutaneous adipose tissue, namely palmitoleic acid, was reduced in Berkshires vs. Ningxiang pigs (Figure 9B). However, KO00061 was not significantly enriched in the comparison between Ningxiang pigs and F_1_ offspring, and between Berkshires and F_1_ offspring, indicating after crossing Berkshires with Ningxiang pigs, DEGs related to the KO00061 pathway (*FASN*, *ACACA*, *CBR4*, *MCH*) of F_1_ offspring were upregulated and the relative content of palmitoleic acid was reduced in the subcutaneous adipose tissue of F_1_ offspring compared to that of Berkshires. 

KO01040, the biosynthesis of unsaturated fatty acids pathway, was enriched in Berkshires vs. Ningxiang pigs. The DEGs related to KO01040, such as elongation of very long chain fatty acids protein 6 (*ELOV6*) and stearoyl-CoA desaturase (*SCD*) were upregulated, and very long-chain (3R)-3-hydroxyacyl-CoA dehydratase 2 (*HACD2*) was downregulated in the subcutaneous adipose tissue of Ningxiang pigs compared to that of Berkshires (Figure 10A). In addition, the content of the SCLs related to kO01040, such as palmitoleic acid, linoleic acid, icosadienoic acid, and arachidonic acid were reduced in the subcutaneous adipose tissue of Berkshires vs. Ningxiang pigs (Figure 10B). The KO00591 (linoleic acid metabolism) pathway was significantly enriched in Berkshires vs. Ningxiang pigs. There was one DEG, namely cytochrome P450 family 3 subfamily A46 (*CYP3A46*), and three SCLs, namely linoleic acid, arachidonic acid and 9,10-DHOME, that were involved in the linoleic acid metabolism pathway. *CYP3A46* was upregulated in the subcutaneous adipose tissue of Ningxiang pigs compared to that of Berkshires.

In KO00590 (arachidonic acid metabolism), the DEGs, namely gamma-glutamyltranspeptidase (*GGT5*), cytochrome P450 4F6-like (*LOC110255237*), microsomal prostaglandin-E synthase 1 (*PTGES*), salivary lipocalin (*SAL1*), and glutathione peroxidase 1 (*GPX1*), were upregulated, while cytochrome P450 family 2 subfamily B22 (*CYP2B22*) and glutathione peroxidase 3 (*GPX3*) were downregulated in the subcutaneous adipose tissue of Ningxiang pigs compared to that of Berkshires (Figure 11A). The relative content of arachidonic acid in the subcutaneous adipose tissue of Ningxiang pigs was lower compared to that of Berkshires. In Berkshires vs. F_1_ offspring, the DEGs related to the KO00590 pathway, namely glutathione peroxidase 1 (*GPX1*), prostaglandin-endoperoxide synthase 2 (*PTGS2*), and phospholipase A2 group IVA (*PLA2G4A*), were upregulated (Figure 11B). The correlation analysis of the SCLs and DEGs in the KO00590 pathway suggested that the expression of *PTGES* and *LOC110255237* were positively correlated with the content of arachidonic acid and some phosphatidylcholines, and the expression of *GPX1* was negatively correlated with the content of arachidonic acid and some phosphatidylcholines in Berkshires vs. Ningxiang pigs (Figure 11C). Moreover, the expression of *GPX1* was negatively correlated with the content of some phosphatidylcholines and the expression of *PLA2G4A* was negatively correlated with the content of one phosphatidylcholine in Berkshires pigs vs. F_1_ offspring (Figure 11D). 

The results indicated that the content of palmitoleic acid, arachidonic acid, linoleic acid, and eicosadienoic acid in the subcutaneous adipose tissue of Ningxiang pigs was lower than that of Berkshires, and the key genes of the KEGG pathways belonging to fatty acid biosynthesis and metabolism exhibited differences in their expression in the subcutaneous adipose tissue between Berkshires and Ningxiang pigs and between Berkshires and F_1_ offspring. The identified key genes may contribute to the improvements in the content of palmitoleic acid, arachidonic acid, linoleic acid, and eicosadienoic acid in Ningxiang pig through molecular design breeding.

For Berkshires vs. Ningxiang pigs, the DEGs related to KO00561 (glycerolipid metabolism) pathways, such as acylglycerol lipase (*MGLL*) and diacylglycerol O-acyltransferase 2 (*DGAT2*) were upregulated, whereas, glycerol kinase (*GK*), diacylglycerol kinase beta (*DGKB*), and phosphatidate phosphatase (*LPIN1*) were downregulated (Figure 12A). The correlation analysis of SCLs and DEGs in the KO00561 pathway suggested that the expression of *DGAT2* was negatively correlated with the content of the majority of diacylglycerols and triglycerides and the expression of *LPIN1* and *DGKB* was positively correlated with the content of the majority of diacylglycerols and triglycerides. Additionally, the expression of *MGLL* was negatively correlated with the content of the majority of diacylglycerols, triglycerides, and phosphatidic acids (Figure 12B).

### 3.6. Validation of the Results by RT-qPCR

To verify the accuracy of RNA-seq data, 12 DEGs (*GPX1*, *GPX3*, *LPIN1*, *MGLL*, *CYP2B22*, *DGAT2*, *FASN*, *ACACA*, *SCD*, *PSPH*, *ELOV6*, and *HACD2*) from the group of Berkshires vs. Ningxiang pigs were randomly selected for RT-qPCR analysis. The results showed that the expression patterns of these DEGs in RT-qPCR were consistent with RNA-seq (Figure 13A), and the correlation between the two methods was relatively strong, with a correlation coefficient of 0.94 (R^2^ = 0.89, Figure 13B), which indicated that the DEGs identified from RNA-seq in this study were reliable.

## 4. Discussion

### 4.1. Large Differences Existed in the Lipid Composition of the Subcutaneous Adipose Tissue among Ningxiang Pigs, Berkshires and F_1_ Offspring

In total, 1293 lipids were detected. Of the detected lipids, 32.48% were TGs, 6.03% were DGs, 6.19% were Cer-ASs, 6.11% were PEs, 5.88% were PCs, and 5.34% were PSs. Compared with Berkshires, Ningxiang pigs had 412 downregulated and 11 upregulated lipids, Cer(d18:0/18:0) was highly significantly upregulated, while linolenic acid and pentadecanoic acid were highly significantly downregulated (Appendix A). Correlation analysis was performed on the SCLs, mapping the top 50 VIP values, in the subcutaneous adipose tissue of Berkshires vs. Ningxiang pigs, and the majority of the lipids were positively correlated, although the Cer(d18:0/18:0) and TG(18:0_16:1_24:6) were negatively correlated (Appendix A). The F_1_ offspring had 251 downregulated and one SP upregulated lipids in the subcutaneous adipose tissue compared to that of Berkshires. Cer(d18:0/18:0) was highly significantly upregulated, while linolenic acid and pentadecanoic acid were not significantly downregulated in the subcutaneous adipose tissue of F_1_ offspring compared to that of Berkshires (Appendix A). For the comparison of Ningxiang pigs vs. F_1_ offspring, 50 SCLs were screened. Compared with Ningxiang pigs, F_1_ offspring had just seven downregulated and 43 upregulated lipids in the subcutaneous adipose tissue, and linolenic acid and pentadecanoic acid were highly significantly downregulated, while Cer(d18:0/18:0) was not significantly upregulated (Appendix A). The relative content of linolenic acid and pentadecanoic acid were highly significantly regulated in the subcutaneous adipose tissue of F_1_ offspring compared to that of Ningxiang pigs.

### 4.2. The DEGs and SCLs Related to the KEGG Pathways Belonging to Fatty Acids Biosynthesis and Metabolism Contributed to the Differences in Fatty Acid Composition of Adipose Tissue and Fat Deposition among Ningxiang, Berkshires, and F_1_ Offspring

The Ningxiang pig is a famous Chinese indigenous breed with excellent meat quality. The Berkshires is a typical lean breed, famous for its high lean meat percentage and fast growth rate. Moreover, Berkshires have been under intensive artificial selection and genetic improvement of important traits, such as growth rate, lean meat percentage and backfat thickness [22]. As knowledge about a healthy diet has grown, the consumers now prefer to increase their intake of PUFAs and reduce their SFAs [23]. Thus, we focused on backfat thickness and the fatty acid composition of subcutaneous adipose tissue. Muti-omics association analyses of local pig breeds and modern pig breeds can screen the candidate genes for backfat thickness and the fatty acid composition of the subcutaneous adipose tissue. These genes likely have undergone artificial selection. In this study, a total of 38 fatty acyls (14 SFAs, 13 MUFAs, and 11 PUFAs) were detected. Two SFAs (C(14:0), pentadecanoic acid), three MUFAs (9,10-DiHOME, palmitoleic acid, and heptadecanoic acid), and five PUFAs (linoleic acid, linolenic acid, eicosadienoic acid, arachidonic acid, and docosapentaenoic acid) were downregulated in the subcutaneous adipose tissue of Berkshires vs. Ningxiang pigs. Moreover, 9,10-DiHOME, linoleic acid and docosapentaenoic acid were downregulated in the subcutaneous adipose tissue of Berkshires vs. F_1_ offspring. In addition, C(14:0), pentadecanoic acid and linolenic acid were upregulated in the subcutaneous adipose tissue of the group Ningxiang pigs vs. F_1_ offspring. This suggested that the relative content of MUFAs and PUFAs in the subcutaneous fat of Berkshires was higher compared to that of Ningxiang pigs or F_1_ offspring, and the relative content of C(14:0), pentadecanoic acid, and linolenic acid of F_1_ offspring was higher compared to that of Ningxiang pigs. It was reported that the adipose tissue of local pig breeds contained a lower proportion of PUFAs compared with that of modern pig breeds [4,24]. This was consistent with our study. Interestingly, these MUFAs (palmitoleic acid) and PUFAs (linolenic acid, linoleic acid, arachidonic acid, and eicosadienoic acid) were involved in multiple KEGG pathways belonging to fatty acids biosynthesis and metabolism, such as the KO00061 (fatty acid biosynthesis) pathway, the KO01040 (biosynthesis of unsaturated fatty acids) pathway, the KO00590 (arachidonic acid metabolism) pathway, and KO00591 (linoleic acid metabolism) pathways. Moreover, DEGs, such as *FASN*, *ACACA*, *CBR4*, and *MCH*, *ELOV6*, *SCD*, and *HACD2, GGT5*, *PTGES*, *SAL1*, *GPX1*, *GPX3*, *CYP2B22*, *PTGS2*, *PLA2G4A,* and *CYP3A46*, may play vital roles in the fatty acid composition of adipose tissue and the content of MUFAs and PUFAs. *FASN* is a key enzyme of the de novo synthesis of fatty acids pathways. *ACACA* catalyzes the irreversible formation of malonyl-CoA from acetyl-CoA, and acetyl-CoA is the precursor of de novo synthesized fatty acids. These genes played important roles in the fatty acid composition of meat [25]. *ELOV6*, *FASN*, and *SCD* were identified by genome-wide association studies (GWAS) as candidate genes for the fatty acid composition of the backfat of multiple pig populations [26,27]. In the Alentejano fat breed, the activity of acetyl-CoA carboxylase in the dorsal subcutaneous fat was three- and ninefold higher than that in the Large White lean breed [28], which was consistent with the results of this study. Moreover, several comparative transcriptomic studies indicated that *FASN* and *ACACA* were upregulated in indigenous breeds compared with modern pig breeds [29,30,31]. *FASN* and *ACACA* play vital roles in fatty acid biosynthesis and lipid metabolism. Moreover, it was reported that the content of palmitoleic acid was closely associated with intramuscular fat, meat color, and PH24h (PH value at 24 h after slaughter), and palmitoleic acid might be a promising indicator of superior meat quality [32]. In this study, the relative content of palmitoleic acid in the subcutaneous adipose tissue of Ningxiang pigs was lower compared to that of Berkshires. *ELOV6* catalyzes the elongation of fatty acids [33], and *SCD* is essential for the desaturation of fatty acids [34]; both *ELOV6* and *SCD* play important roles in the synthesis of MUFAs. It has been reported that Meishan pigs show lower activity of stearoyl-coA desaturase in the subcutaneous adipose tissue compared to that of Large White [15]. This was also consistent with our study. In this study, *SCD* was upregulated in the subcutaneous adipose tissue for the group of Berkshires vs. Ningxiang pigs. These DEGs and SCLs contributed to the differences in the fatty acid composition of adipose tissue and fat deposition among Ningxiang pig, Berkshires, and F_1_ offspring.

### 4.3. Overlap between KEGG Pathway-Related Genes Screened by Muti-Omics Association Analysis and Artificially Selected Genes

The key genes screened by muti-omics association analysis were probably selected by humans when breeding pigs. The analysis revealed 19, 16, and seven KEGG pathways (*p* < 0.05) enriched in Berkshires vs. Ningxiang pigs, in Berkshires vs. F_1_ offspring, and in Ningxiang pigs vs. F_1_ offspring, respectively, according to the transcriptomics data (Figure 7B). Additionally, 52, 30, and 35 KEGG pathways (*p* < 0.05) were enriched in Berkshires vs. Ningxiang pigs, in Berkshires vs. F_1_ offspring, and in Ningxiang pigs vs. F_1_ offspring, respectively, according to the transcriptomics data (Figure 4). The comparative genomic analysis between indigenous breeds and modern pig breeds uncovered the selection regions and KEGG pathways related to the candidate genes in those regions [35,36,37,38]. A comparative genomic analysis indicated that the candidate genes related to KEGG pathway ‘phosphatidylinositol signaling system’, and ‘inositol phosphate metabolism’ in the selection regions indicated by the Fst statistics were significantly enriched in Duroc vs. Xiang pig, and the candidate genes related to the KEGG pathway ‘thermogenesis’ in selection regions indicated by the Fst statistics were enriched in Yorkshire vs. Xiang pigs [39]. Another study found that the KEGG pathway ‘carbohydrate digestion and absorption’ related to the candidate genes in selective regions from Fst statistics was significantly enriched in the group of Anqing six-end-white pig (ASP) vs. Asian wild boars [40]. Interestingly, the results of our study suggested that the KO04070 (phosphatidylinositol signaling system) pathway, the KO00562 (inositol phosphate metabolism) pathway, and KO04714 (thermogenesis) pathways were significantly enriched in Berkshires vs. Ningxiang pigs and in Berkshires vs. F_1_ offspring. The KO04973 (carbohydrate digestion and absorption) pathway was significantly enriched in the group of Berkshires vs. Ningxiang pigs. This suggested that the genes related to these KEGG pathways were probably under intense artificial selection. For instance, *FASN* and *ACACA* were the DEGs of the KO00061 (fatty acid biosynthesis) pathway, which was significantly enriched in the groups of Berkshires vs. Ningxiang pigs. The Asian allele of *FASN* was thought to have an important effect on backfat phenotypes and was under strong selection in modern European pig breeds [41]. *ACACA* is located in selected regions detected by the CLR test in Chinese indigenous breeds and Yorkshires [42].

### 4.4. Application of Multi-Omics Data to Pig Breeding for Improving the Backfat Thickness and Fatty Acid Composition of Adipose Tissue

With the dramatic growth of multi-omics data, its integration into analyses can reveal candidate genes for complex traits and gene regulatory networks. However, it is still a challenge to integrate the multi-omics data [43]. The identification of functional variants in candidate genes, which can be screened through the use of integrated multi-omics data, may provide more useful new functional molecular markers and biomarkers for pig breeding [23,44,45]. Intelligent algorithms were used to optimize the prediction effect of these functional molecular markers and biomarkers [46]. Thus, mining the candidate genes using multi-omics data was an important step. In this study, multiple genes related to KEGG pathways were associated with backfat thickness and the fatty acid composition of adipose tissue. Some genes, such as *FASN*, *SCD*, *ACACA*, and *ELOV6*, are considered to be important functional genes for backfat thickness and the fatty acid composition of adipose tissue. We validated the credibility of our research results in multiple dimensions. These candidate genes will be used to identify functional variants and develop functional molecular markers. We have provided a possible method of applying multi-omics data to pig breeding.

## 5. Conclusions

Ningxiang pig acted as the male parents in a cross with Berkshires, and the meat quality of their offspring (F_1_) was superior (data unpublished). In order to study the molecular mechanism of the differences in fat deposition among Ningxiang, Berkshires, and F_1_ offspring, transcriptomic, and lipidomic analyses were conducted. The functional enrichment analysis indicated that the pathways of the glycine, serine, and threonine metabolism, arachidonic acid metabolism, fatty acid biosynthesis, linoleic acid metabolism, alpha-linolenic acid metabolism, and glycerophospholipid metabolism were significantly enriched in comparisons of the Ningxiang pig, Berkshires, and F_1_ crosses vs. Berkshires. The difference in fat deposition among Ningxiang pig, Berkshires, and F_1_ offspring may be caused by the differences in the expression pattern of the key genes in multiple enrichment KEGG pathways. We found that the DEGs (*PHGDH*, *LOC110256000*) and the SCLs (phosphatidylserines) were significantly correlated and played a vital role in glycine, serine, and threonine metabolism. *FASN*, *ACACA*, *CBR4*, *SCD*, *ELOV6*, *HACD*, *CYP3A46*, *CYP2B22*, *GPX1*, and *GPX3*, which are related to fatty acid biosynthesis and metabolism, were identified as key genes that affect the content of palmitoleic acid, linoleic acid, arachidonic acid, and eicosadienoic acid in the subcutaneous adipose tissue. This study screened key genes and lipids, and investigated the molecular mechanisms of the differences in fat deposition among Ningxiang pig, Berkshires, and F_1_ offspring.

## Figures and Tables

**Figure 1 animals-13-03321-f001:**
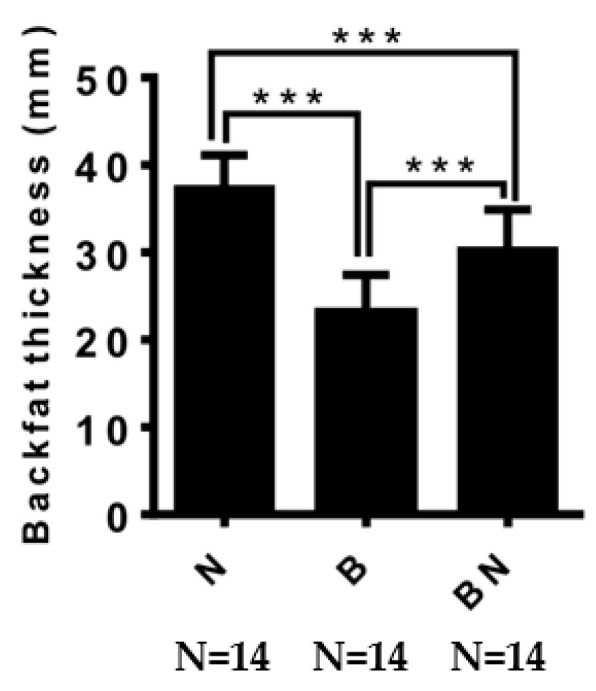
The difference in backfat thickness among Ningxiang pigs, Berkshires, and F_1_ offspring. B, Berkshires; N, Ningxiang pig; BN, F_1_ offspring. *** represent *p* < 0.001.

**Figure 2 animals-13-03321-f002:**
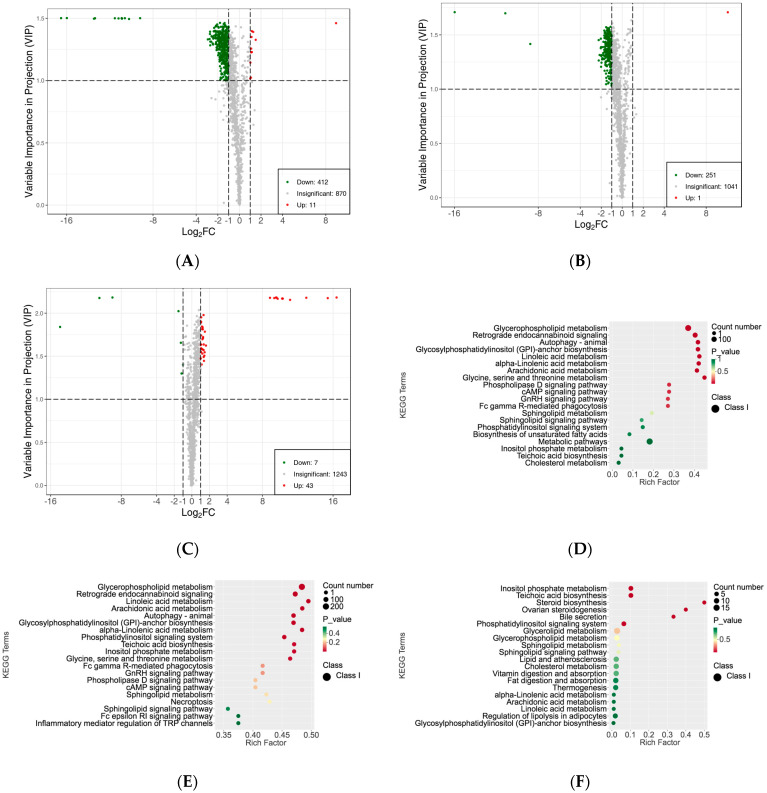
Differences in the lipidomic profiles of the subcutaneous adipose tissue. (**A**–**C**) Volcano plots of SCLs for the subcutaneous adipose tissue in Berkshires vs. Ningxiang pigs (**A**), Berkshires vs. F_1_ offspring (**B**), Ningxiang pigs vs. F_1_ offspring (**C**), green dots represent downregulated SCLs, red dots represent upregulated SCLs. (**D**–**F**) Diagrams of the degree of KEGG pathway enrichment of SCLs in the subcutaneous adipose tissue from the group of Berkshires vs. Ningxiang pigs (**D**), Berkshires vs. F_1_ offspring (**E**), and Ningxiang pigs vs. F_1_ offspring (**F**). Eighteen samples of subcutaneous adipose from six Ningxiang pigs, six Berkshires, and six F_1_ pigs, respectively, were used to detect lipidomic profile.

**Figure 3 animals-13-03321-f003:**
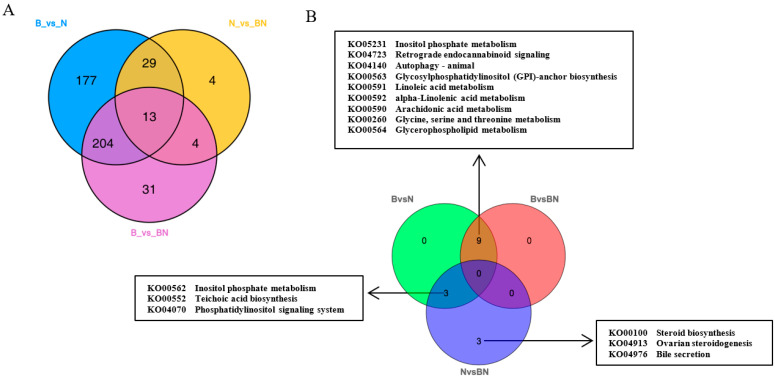
(**A**) Venn diagram of the SCLs identified by lipidomic analysis in the groups of Berkshires vs. Ningxiang pigs, in Berkshires vs. F_1_ offspring, and in Ningxiang pigs vs. F_1_ offspring. (**B**) Venn diagram analysis of KEGG pathways (*p* < 0.05) based on the SCLs in Berkshires vs. Ningxiang pigs, in Berkshires vs. F_1_ offspring, and in Ningxiang pigs vs. F_1_ offspring. Eighteen samples of subcutaneous adipose from six Ningxiang pigs, six Berkshires and six F_1_ pigs, respectively, were used to detect lipidomic profile. B, Berkshires; N, Ningxiang pig; BN, F_1_ offspring.

**Figure 4 animals-13-03321-f004:**
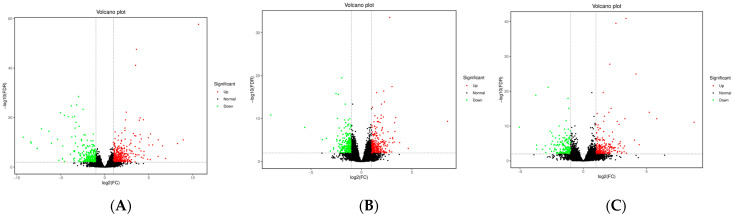
Volcano plots of DEGs for the subcutaneous adipose tissue in the groups of Berkshires vs. Ningxiang pigs (**A**), Ningxiang pigs vs. F_1_ offspring (**B**), and Berkshires vs. F_1_ offspring (**C**). Twelve samples of subcutaneous adipose from four Ningxiang pigs, four Berkshires, and four F_1_ pigs, respectively, were used to detect transcriptomic profile.

**Figure 5 animals-13-03321-f005:**
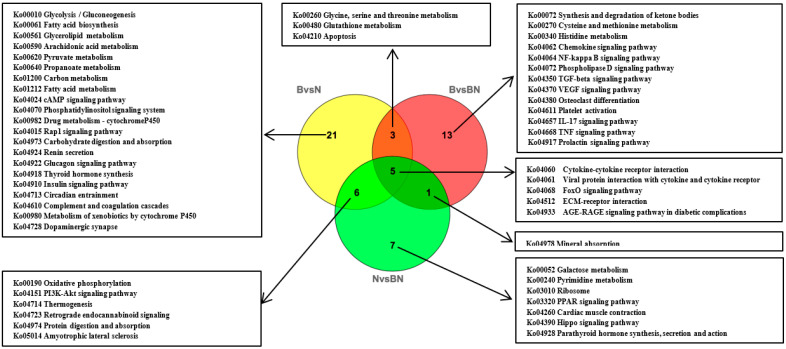
Venn diagram of the KEGG pathway analysis (*p* < 0.05) based on DEGs in Berkshires vs. Ningxiang pigs, F_1_ offspring vs. Berkshires, and Ningxiang pigs vs. F_1_ offspring, based on transcriptomic data. B, Berkshires; N, Ningxiang pigs; BN, F_1_ offspring. Twelve samples of subcutaneous adipose from four Ningxiang pigs, four Berkshires, and four F_1_ pigs, respectively, were used to detect transcriptomic profile.

**Figure 6 animals-13-03321-f006:**
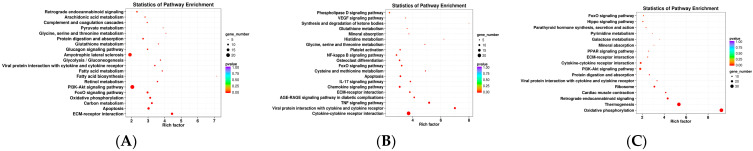
The diagrams for the KEGG pathway enrichment of DEGs in Berkshires vs. Ningxiang pigs (**A**), Ningxiang pigs vs. F_1_ offspring (**B**), and Berkshires vs. F_1_ offspring (**C**). Twelve samples of subcutaneous adipose from four Ningxiang pigs, four Berkshires, and four F_1_ pigs, respectively, were used to detect the transcriptomic profile.

**Figure 7 animals-13-03321-f007:**
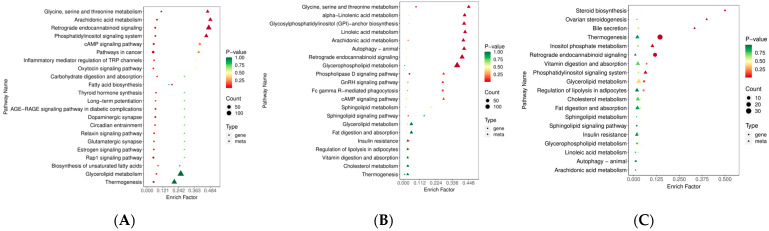
The diagrams for the degree of KEGG pathway enrichment of the DEGs and SCLs in the subcutaneous adipose tissue from Berkshires vs. Ningxiang pigs (**A**), Berkshires vs. F_1_ offspring (**B**), Ningxiang pigs vs. F_1_ offspring (**C**).

**Figure 8 animals-13-03321-f008:**
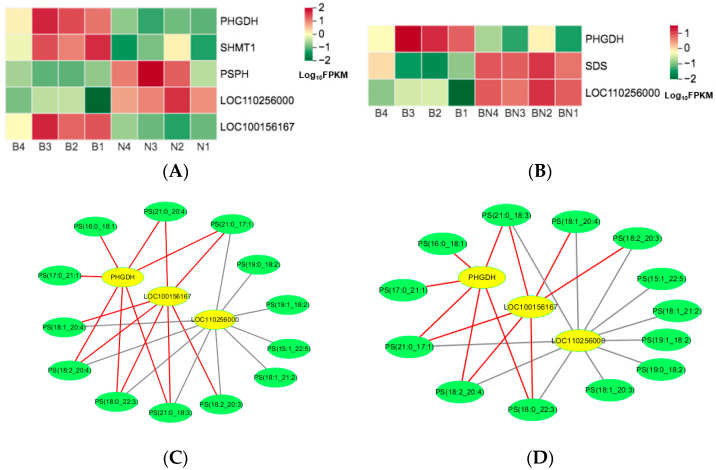
Heat maps of DEGs for the KO00260 pathway in Berkshires vs. Ningxiang pigs (**A**) and Berkshires vs. F_1_ offspring (**B**), respectively. B, Berkshires; N, Ningxiang pigs; BN, F_1_ offspring. Correlation network diagram for the KO00260 (the lycine, serine and threonine metabolism) pathways in Berkshires vs. Ningxiang pigs (**C**) and Berkshires vs. F_1_ offspring (**D**), respectively. Yellow circles represent genes, green circles represent lipids, red lines represent positive correlations, gray lines represent negative correlations.

**Figure 9 animals-13-03321-f009:**
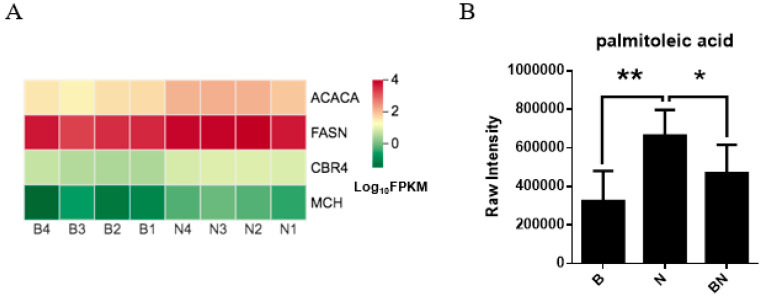
(**A**) Heat maps of the DEGs for the KO00061 (fatty acid biosynthesis) pathways in Berkshires vs. Ningxiang pigs. B, Berkshires; N, Ningxiang pigs. (**B**) The column chart of palmitoleic acid, in Berkshires pigs vs. Ningxiang pigs. B, Berkshires; N, Ningxiang pigs; BN, F_1_ offspring. * represents *p* < 0.05, ** represents *p* < 0.01.

**Figure 10 animals-13-03321-f010:**
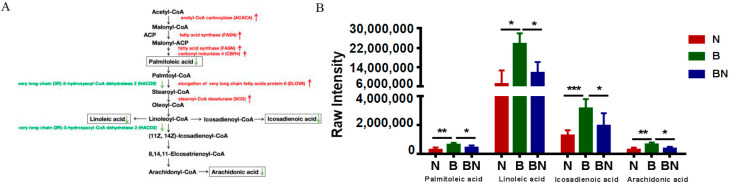
(**A**) The DEGs and SCLs related to the KO01040 (biosynthesis of unsaturated fatty acids) in Berkshires vs. Ningxiang pigs. Red represents upregulation, green represents downregulation. (**B**) The column chart of the SCLs related to the KEGG pathway (kO01040) in Berkshires, Ningxiang pigs, and F_1_ pigs. * represents *p* < 0.05, ** represents *p* < 0.01, *** represents *p* < 0.001.

**Figure 11 animals-13-03321-f011:**
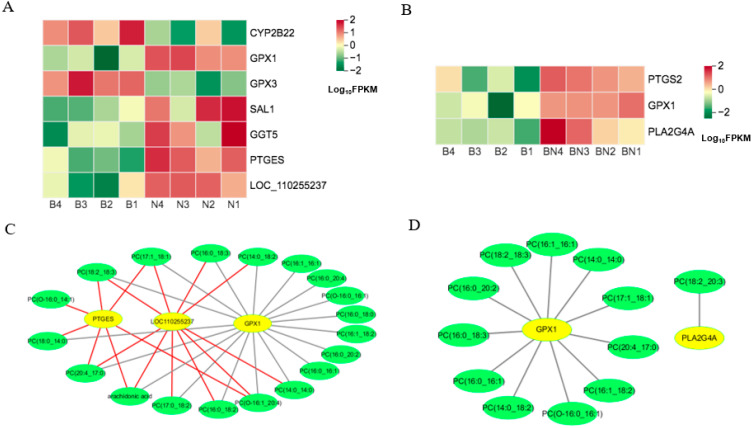
(**A**,**B**) Heat maps of the DEGs for the KO00590 pathway in Berkshires vs. Ningxiang pigs (**A**) and Berkshires vs. F_1_ offspring (**B**). B, Berkshires; N, Ningxiang pigs; BN, F_1_ offspring. Correlation network diagram for the KO00590 pathway in Berkshires vs. Ningxiang pigs (**C**) and Berkshires vs. F_1_ offspring (**D**). Yellow circles represent genes, green circles represent lipids, red lines represent positive correlations and gray lines represent negative correlations.

**Figure 12 animals-13-03321-f012:**
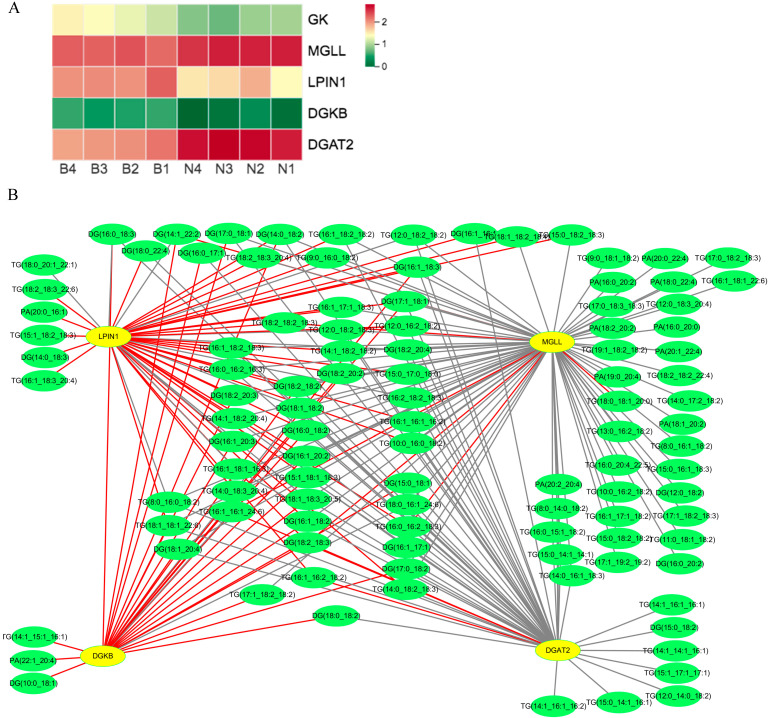
(**A**) Heat maps of the DEGs for the KO00561 pathway in Berkshires vs. Ningxiang pigs. B, Berkshires; N, Ningxiang pigs; BN, F_1_ offspring. (**B**) Correlation network diagram for the KO00561 pathway in Berkshires vs. Ningxiang pigs. Yellow circles represent genes, green circles represent lipids, red lines represent positive correlations, and gray lines represent negative correlations.

**Figure 13 animals-13-03321-f013:**
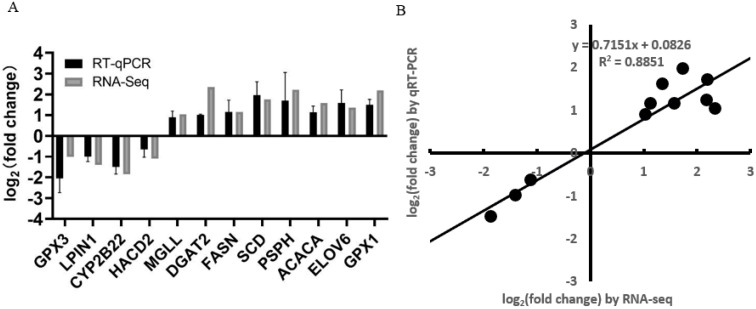
(**A**) Histogram of RNA-seq and RT−qPCR expression levels. The X−axis represents the 12 selected DEGs, and the Y-axis represents the expression levels of DEGs from RNA−seq and RT−qPCR. (**B**) The linear regression analysis of expression level between RNA−seq and RT−qPCR data. Eighteen samples of subcutaneous adipose from six Ningxiang pigs, six Berkshires and six F_1_ pigs, respectively, were used to conduct RT−qPCR.

## Data Availability

The data presented in this study are available in the Appendix A.

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
