# Peer review of "Integrative Analysis of Transcriptomic and Lipidomic Profiles Reveals a Differential Subcutaneous Adipose Tissue Mechanism among Ningxiang Pig and Berkshires, and Their Offspring"

_animals, 2023, doi:10.3390/ani13213321_

Round 1

Reviewer 1 Report (Previous Reviewer 3)

The authors have adequately addressed my concerns in this version of the manuscript and have acknowledged the limitations of the study.

Author Response

Response to Reviewer 1 Comments

The authors have adequately addressed my concerns in this version of the manuscript and have acknowledged the limitations of the study.

Response:  Thank you for the review's comments,your comments plays vital role in improving this manuscript. Because of limitations of the small sample sizes (the sample size of 4 samples from each genetic grouping may not sufficient for the analysis performed), the genes identified as DEGs could have occurred by chance, the results of the transcriptome data analysis are speculative. This article revealed multiple lipids that are potentially available biological indicators and screened key genes that play vital role in the fat deposition. We still acknowledged the limitations of the study.

Reviewer 2 Report (New Reviewer)

Major concerns:

1.      The authors escaped the genetic factor in deciding the fat deposition. This question is critical and cannot avoided considering the authors comparing Ningxiang pig, .Berkshire pig and their hybrids?

Minor issues:

1.      The article needed to be reorganized as the main results are based only on the transcriptomic and lipidomic lines, which can briefly presented instead of long text describing.

2.      Plenty of highlighted background in text, which greatly affects reading.

3.      The figures are bad organized and in low quality, for example,Fig2, barely to see , Fig3, some text in figures are even hided or incompletely displayed.

Author Response

Response to Reviewer 2 Comments

Major concerns:

1.  The authors escaped the genetic factor in deciding the fat deposition. This question is critical and cannot avoided considering the authors comparing Ningxiang pig, .Berkshire pig and their hybrids?

Response:  Thank you for the review's comments. Adipose tissue composition contributes greatly to the quality and nutritional value of meat, the obvious phenotype differences existed in the subcutaneous adipose tissue between Chinese local pig breeds and modern pig breeds. This study reveal differential mechanism of subcutaneous adipose tissue among Ningxiang pigs, Berkshires and their offspring through integrative analysis of transcriptomic and lipidomic profiles. The genetic factor in deciding the fat deposition is complicated. We screened key genes and lipids which play vital role in the fat deposition and investigated the molecular mechanisms of the differences in fat deposition through comparing Ningxiang pig, Berkshires and their hybrids. We found that FASN, ACACA, CBR4, MCH, ELOV6, SCD, HACD2, GGT5, PTGES, SAL1, GPX1, GPX3, CYP2B22, PTGS2, PLA2G4A, may play vital roles in the fatty acid composition of adipose tissue and the content of MUFAs and PUFAs. Ningxiang pig show lower activity of stearoyl-coA desaturase, and higher activity of acetyl-CoA carboxylase in the subcutaneous adipose tissue compared to Berkshires. The content of palmitoleic acid, arachidonic acid, linoleic acid, and eicosadienoic acid in the subcutaneous adipose tissue of Ningxiang pigs was lower than that of Berkshires, and the key genes of the KEGG pathways belonging to fatty acids biosynthesis and metabolism exhibited differences in their expression in the subcutaneous adipose tissue between Berkshires and Ningxiang pigs and between Berkshires and F1 offspring. Besides, DGAT2, MGLL, GK, LPIN1, play vital role in glycerolipid metabolism, and glycerolipid was important component of subcutaneous adipose tissue. Those genes screened in this study may affect the fat deposition. So we screened key genes of multiple KEGG pathway, which may decide the fat deposition.

Minor issues:

  1. The article needed to be reorganized as the main results are based only on the transcriptomic and lipidomic lines, which can briefly presented instead of long text describing.

Response:  Thank you for the review's comments. your comments play vital role in improving this manuscript. Lycine, serine and threonine metabolism pathway, fatty acid biosynthesis and metabolism related pathways and glycerolipid metabolism pathway were siginificant enriched in comparisons of Berkshire vs. Ningxiang pigs and Berkshires vs. F1 offspring through integrative analysis of transcriptomic and lipidomic profiles. Nest, we screened key genes of the KEGG pathway, which may decide the fat deposition, and screened the multiple lipids that are potentially available biological indicators.

  1. Plenty of highlighted background in text, which greatly affects reading.

Response:  Thank you for the review's comments,your comments plays vital role in improving this manuscript. We have corrected in the revised manuscript.

    3.  The figures are bad organized and in low quality, for example,Fig2, barely to see , Fig 3, some text in figures are even hided or incompletely displayed.

Response:  Thank you for the review's comments,your comments plays vital role in improving this manuscript. We have organized the figures again, we have redrawn Figure 2, Figure 7 and enlarged the annotations,We organized the figure 3 again, the text in figures are completely displayed. We also organized the figure 1, figure 9, figure 10 again.

Reviewer 3 Report (New Reviewer)

This manuscript used transcriptomic and lipidomic profiles to study the molecular mechanisms of the difference in fat deposition among Ningxiang pig, Berkshire and F1 offspring, and to identify the key genes and lipidsIt provided insights into selection for backfat thickness and the fat composition of adipose tissue for pig breeding.

However, the manuscript still needs revision before the acceptance for publication.

Comments:

1. Total 42 pigs (14 per group) from the same farm were used in this study. Why only 12 samples of subcutaneous adipose from 12 pigs (4 per group) were used to  detect the transcriptomic profiles, and 18 samples of subcutaneous adipose were used to detect the lipidomic profiles? The samples used for transcriptomic profiles and lipidomic profiles were different. Please interpret the reason.

2. The numbers of sample in each figure should be supplemented. 

Minor editing of English language required.

Author Response

Response to Reviewer 2 Comments

This manuscript used transcriptomic and lipidomic profiles to study the molecular mechanisms of the difference in fat deposition among Ningxiang pig, Berkshire and F1 offspring, and to identify the key genes and lipids. It provided insights into selection for backfat thickness and the fat composition of adipose tissue for pig breeding.

However, the manuscript still needs revision before the acceptance for publication.

Comments:

  1. Total 42 pigs (14 per group) from the same farm were used in this study. Why only 12 samples of subcutaneous adipose from 12 pigs (4 per group) were used to  detect the transcriptomic profiles, and 18 samples of subcutaneous adipose were used to detect the lipidomic profiles? The samples used for transcriptomic profiles and lipidomic profiles were different. Please interpret the reason.

Response:  Thank you for the review's comments. Due to research funding limitations,we selected 12 samples of subcutaneous adipose from 4 Ningxiang pigs, 4 Berkshire pigs and 4 F1 pigs, respectively, to detect transcriptomic profile. For lipidomic profile, in order to make this research more valuable,we still added some samples even with limited funding after referring to relevant articles. Finally, we selected 18 samples of subcutaneous adipose from 6 Ningxiang pigs, 6 Berkshire pigs and 6 F1 pigs, respectively, to detect lipidomic profile. We also conduct RT-qPCR validation.

  1. The numbers of sample in each figure should be supplemented.

Response:  Thank you for the review's comments,your comments plays vital role in improving this manuscript. We have added the numbers of sample in the figures or the annotations of the figure.

This manuscript is a resubmission of an earlier submission. The following is a list of the peer review reports and author responses from that submission.

Round 1

Reviewer 1 Report

Adipose tissue composition contributes greatly to the quality and nutritional value of meat. The obvious phenotype differences existed in the subcutaneous adipose tissue between Chinese local pig breeds and modern pig breeds. This study reveal differential mechanism of subcutaneous adipose tissue among Ningxiang, Berkshire and their offspring pigs through integrative analysis of transcriptomic and lipidomic profiles. The results obtained as part of the research indicate a detailed, extensive knowledge by the authors of not only the subject but also the dependencies resulting from their interpretation. The presented manuscript meets the requirements of the original scientific publication and can be published after minor changes. However, I would recommend that the authors pay particular attention to some sections. I have some comments about this work:

(1) In figure 6, Chinese character need delete.

(2) In Table S1, DEGs in the groups of Berkshire pigs vs Ningxiang pigs enriched in partial KEGG pathways. For glycerolipid metabolism pathway. the FDR of DGAT2 reach 8.38E-20, Which indicated that this gene may extreme differential expression between Berkshire pigs vs Ningxiang pigs. For glycerolipid metabolism, authors are suggest to further analysis.

(3) Line 671-674, Replace Samples into pigs.

(4) Line 678-679, Replace P-value into p-value.

(5) Line 735-736, “We found that the DEGs (PHGDH, AOC2) and the SCLs (Phosphatidylserines) were significantly correlated and play vital role to lycine, serine and threonine metabolism”, which was not consistent with the abstract. Replace AOC2 into LOC110256000.

(6) Line 737-738, “FASN, ACACA, CBR4, SCD, ELOV6, HACD, CYP3A, CYP2B, gpx belonging to fatty acid biosynthesis and metabolism were identified as key genes”, which was not consistent with the abstract. Replace CYP3A, CYP2B, gpx into CYP3A46, CYP2B22, GPX1, GPX3.

(7) Authors need to add the ethical statement in the manuscript.

Author Response

Response to Reviewer 1 Comments

Adipose tissue composition contributes greatly to the quality and nutritional value of meat. The obvious phenotype differences existed in the subcutaneous adipose tissue between Chinese local pig breeds and modern pig breeds. This study reveal differential mechanism of subcutaneous adipose tissue among Ningxiang, Berkshire and their offspring pigs through integrative analysis of transcriptomic and lipidomic profiles. The results obtained as part of the research indicate a detailed, extensive knowledge by the authors of not only the subject but also the dependencies resulting from their interpretation. The presented manuscript meets the requirements of the original scientific publication and can be published after minor changes. However, I would recommend that the authors pay particular attention to some sections. I have some comments about this work:

(1) In figure 6, Chinese character need delete.

Response 1: The Chinese character had been deleted in the revised manuscript.

(2) In Table S1, DEGs in the groups of Berkshire pigs vs Ningxiang pigs enriched in partial KEGG pathways. For glycerolipid metabolism pathway. the FDR of DGAT2 reach 8.38E-20, Which indicated that this gene may extreme differential expression between Berkshire pigs vs Ningxiang pigs. For glycerolipid metabolism, authors are suggest to further analysis.

Response 2: We have analysis the glycerolipid metabolism pathway in the revised manuscript.

(3) Line 671-674, Replace Samples into pigs.

Response 3: It have been corrected in the revised manuscript.

(4) Line 678-679, Replace P-value into p-value.

Response 4: It have been corrected in the revised manuscript.

(5) Line 735-736, “We found that the DEGs (PHGDH, AOC2) and the SCLs (Phosphatidylserines) were significantly correlated and play vital role to lycine, serine and threonine metabolism”, which was not consistent with the abstract. Replace AOC2 into LOC110256000.

Response 5: We replace AOC2 into LOC11025600 in the revised manuscript.

(6) Line 737-738, “FASN, ACACA, CBR4, SCD, ELOV6, HACD, CYP3A, CYP2B, gpx belonging to fatty acid biosynthesis and metabolism were identified as key genes”, which was not consistent with the abstract. Replace CYP3A, CYP2B, gpx into CYP3A46, CYP2B22, GPX1, GPX3.

Response 5: We replace CYP3A, CYP2B, gpx into CYP3A46, CYP2B22, GPX1, GPX3 in the revised manuscript.

(7) Authors need to add the ethical statement in the manuscript.

Response 7: We have add the ethical statement in the revised manuscript.

Reviewer 2 Report

Manuscript

Integrative Analysis of Transcriptomic and Lipidomic Profiles 2 Reveal Differential Mechanism of Subcutaneous Adipose 3 Tissue among Ningxiang, Berkshire and Their Offspring Pigs

The manuscript shows an interesting approach to integrating transcriptome data with lipidomics analysis. It is still a novel approach to the problem of animal breeding, where molecular biology can help answer many questions in the field of lipid metabolism regulation and be in future applied "in fields" through proper animal selection. Unfortunately manuscript shows a few flaws [described below] that force me to deny publication of this work. I hope authors overcome disadvantages [especially in point 1] and publish their scientific results.

1]

RnaSEQ results lack method validation. It is hard to consider lipidomics as so. A usually most common method would be real-time PCR with a few most prominent differentially expressed genes. I would suggest that the authors validate the data and add results to the manuscript. Without it, it is challenging to approach data with confidence. The analysis gave exciting results. Together with lipidomics data, it is an interesting approach and invalidated RNAseq data is difficult to publish. That is the most serious flaw in this otherwise good manuscript, and I hope it can be improved.

2]

What was the rationale behind the results cutoff, that is, "|log2Foldchange| > 1, and 126 false discovery rate (FDR) < 0.05"?

3]

Koba's software that is used for enrichment tests is used for KEGG's gene enrichment pathways and is, as far as I know, dedicated to human and mouse genomes, which is a frequent inconvenience when analyzing farm animals' genomes. Although on the gene set level, when operating common gene names, that approach is prone to many mistakes coming from different annotations of genes between specimens, as well as imbalance in deep genome knowledge, that include known splicing variants and relations between genes. There is the possibility to perform gene enrichment solely on the pig genome; for example, many available plugins for Cytoscape, where authors could build up gene networks based solely on the sus scrofa genome. I am not negating the presented approach, but I would like to ask for more explanation.

4]

Another question is about a small-sample group. In lines 126-131, it is stated that RNA-seq analysis was done on three 4-specimen groups. Usually, in publications, can be found much bigger samples within the group, as well as a low amount of 6 per group. The too-small groups raise concerns about the "size" of differences between groups that can be found significant, as well as the representativeness of RNA-seq results. Could authors comment on group size influence their results?

Kind regards,

Author Response

Reviewer 2

Integrative Analysis of Transcriptomic and Lipidomic Profiles 2 Reveal Differential Mechanism of Subcutaneous Adipose 3 Tissue among Ningxiang, Berkshire and Their Offspring Pigs

The manuscript shows an interesting approach to integrating transcriptome data with lipidomics analysis. It is still a novel approach to the problem of animal breeding, where molecular biology can help answer many questions in the field of lipid metabolism regulation and be in future applied "in fields" through proper animal selection. Unfortunately manuscript shows a few flaws [described below] that force me to deny publication of this work. I hope authors overcome disadvantages [especially in point 1] and publish their scientific results.

(1) RnaSEQ results lack method validation. It is hard to consider lipidomics as so. A usually most common method would be real-time PCR with a few most prominent differentially expressed genes. I would suggest that the authors validate the data and add results to the manuscript. Without it, it is challenging to approach data with confidence. The analysis gave exciting results. Together with lipidomics data, it is an interesting approach and invalidated RNAseq data is difficult to publish. That is the most serious flaw in this otherwise good manuscript, and I hope it can be improved.

Response 1: We have added the validation experiment in the revised manuscript. To verify the accuracy of RNA-seq data, eleven DEGs (GPX1, GPX3, LPIN1, MGLL, CYP2B22, DGAT2, FASN, ACACA, SCD, PSPH, ELOV6, HACD2) from the group of Berkshire pigs vs Ningxiang pigs were randomly selected for RT-PCR analysis. The results showed that the expression patterns of these DEGs in qRT-PCR were consistent with RNA-seq.

(2) What was the rationale behind the results cutoff, that is, "|log2Foldchange| > 1, and 126 false discovery rate (FDR) < 0.05"?

Response 2: Foldchange represents the ratio of expression levels between two samples or groups. Generally, the selection criteria for differentially expressed genes (DEGs) are  |log2Foldchange| > 1; Because the differential expression analysis of Transcriptome sequencing is an independent statistical hypothesis test for a large number of gene expression values, there will be a false positive problem. Therefore, the recognized Benjamin Hochberg correction method was used to correct the significance p value (p-value) obtained from the original hypothesis test, and finally FDR was used as the key indicator for screening differential expression genes. Generally, FDR< 0.05 is taken as the default standard.

(3) Koba's software that is used for enrichment tests is used for KEGG's gene enrichment pathways and is, as far as I know, dedicated to human and mouse genomes, which is a frequent inconvenience when analyzing farm animals' genomes. Although on the gene set level, when operating common gene names, that approach is prone to many mistakes coming from different annotations of genes between specimens, as well as imbalance in deep genome knowledge, that include known splicing variants and relations between genes. There is the possibility to perform gene enrichment solely on the pig genome; for example, many available plugins for Cytoscape, where authors could build up gene networks based solely on the sus scrofa genome. I am not negating the presented approach, but I would like to ask for more explanation.

Response 3: When analyzing farm animals’ genome, KEGG enrichment analysis was useful to screen vital pathways. We checked the possible mistakes coming from different annotations of genes between specimens, as well as imbalance in deep genome knowledge, and check the common gene names based on sus scrofa genome. In addition, we removed the terms not associated with biological functions in pigs. KEGG enrichment analysis provided accurate and valuable information in this study.

(4) Another question is about a small-sample group. In lines 126-131, it is stated that RNA-seq analysis was done on three 4-specimen groups. Usually, in publications, can be found much bigger samples within the group, as well as a low amount of 6 per group. The too-small groups raise concerns about the "size" of differences between groups that can be found significant, as well as the representativeness of RNA-seq results. Could authors comment on group size influence their results?

Response 4: the sample group size in this study did not influence the results. In some publications, a low amount of 4 per group was also found. RT-qPCR validation  further indicated that a low amount of 4 per group in this study did not influence the results.

Reviewer 3 Report

This paper compares the gene expression and lipid profile from three genetic groups of pigs, Berkshire, native pigs, and an F1 cross pig. The study is interesting but there are a number of issues with this paper. 

First, the overall english needs to be improved. 

The introduction is overlooking some important recent research and should be updated.

RNA quality metrics need to be included. This should include average RIN value and range. 

The sample size of 4 samples from each genetic grouping is barely sufficient for the analysis performed and from the data appears highly variable. A note that interpretation is limited by the sample size and that the genes identified as DE could have occurred by chance. 

The results are complex and much of this could be simplified and be reported in the appendix rather than the paper.  The results should be focused on the most important findings. 

There is almost no discussion of the results and the discussion that is included is superficial. The results and discussion should be rewritten to focus on the most important finding or result. 

The functional enrichments reported need to be sorted and terms not associated with biological functions in pigs need to be removed - example pancreatic cancer is not relevant to pork production and pork health. 

The relevance and application of the important findings need to be better explained and recommendations made.

There are a large number of language editing problems with this manuscript. The word choice and sentence structure overall needs significant improvement.

Author Response

Reviewer 3

This paper compares the gene expression and lipid profile from three genetic groups of pigs, Berkshire, native pigs, and an F1 cross pig. The study is interesting but there are a number of issues with this paper.

(1) First, the overall english needs to be improved.

Response 1: We have improve the overall English. 

(2) The introduction is overlooking some important recent research and should be updated.

Response 2: We have updated three important recent research in the revised manuscript.

(3) RNA quality metrics need to be included. This should include average RIN value and range.

Response 3: Purity, concentration and integrity of RNA sample were examined by Agilent 2100. Only RNA with good quality could move on to following procedures. The average RIN value is 6.8 and the RIN value is vary from 6.2 to 9.3.

(4) The sample size of 4 samples from each genetic grouping is barely sufficient for the analysis performed and from the data appears highly variable. A note that interpretation is limited by the sample size and that the genes identified as DE could have occurred by chance.

Response 4: The variation between varieties is significantly greater than that within species in this study. RT-qPCR validation indicated that a low amount of 4 per group in this study did not influence the results.

(5) The results are complex and much of this could be simplified and be reported in the appendix rather than the paper.  The results should be focused on the most important findings.

Response 5 : We have moved figures 2 to the appendix, and simplified the results in the revised manuscript.

(6) There is almost no discussion of the results and the discussion that is included is superficial. The results and discussion should be rewritten to focus on the most important finding or result.

Response 6: We have added some content and revised the results in the revised manuscript. The discussion discuss the DEGs and SCLs, and try to improve backfat thickness in the field of lipid metabolism regulation through proper animal selection. The discussion also provided an insight into the selection for backfat thickness and the fat composition of adipose tissue for future breeding strategies.

(7) The functional enrichments reported need to be sorted and terms not associated with biological functions in pigs need to be removed - example pancreatic cancer is not relevant to pork production and pork health.

Response 7: We have sorted the functional enrichments and removed the terms not associated with biological functions in pigs.

(8) The relevance and application of the important findings need to be better explained and recommendations made.

Response 8: We explain the important findings and in the discussion, we also explained the the DEGs (FASN, ACACA, CBR4, SCD, ELOV6, HACD2, CYP3A46, CYP2B22, GPX1, and GPX3) and the SCLs (palmitoleic acid, linoleic acid, arachidonic acid and icosadienoic acid). We concluded that The DEGs and SCLs related to the KEGG pathways belonging to fatty acids biosynthesis and metabolism contributed to the differences of fatty acids composition of adipose tissue and fat deposition among Ningxiang, Berkshire and F1 pigs.

(9) There are a large number of language editing problems with this manuscript. The word choice and sentence structure overall needs significant improvement.

Response 9: We have improved the word choice and sentence structure overall in the revised manuscript.

Round 2

Reviewer 2 Report

Dear Authors,

RnaSEQ validation removed the major flaw in the previous version and gave proper  RnaSeq data reference to other techniques.

All the other comments were satisfyingly explained, and I have no questions or remarks whatsoever.

Therefore I recommend the manuscript for publication.

Kind Regards,

Author Response

RnaSEQ validation removed the major flaw in the previous version and gave proper  RnaSeq data reference to other techniques.

All the other comments were satisfyingly explained, and I have no questions or remarks whatsoever.

Therefore I recommend the manuscript for publication.

Response: Reviewer 2 have no questions and recommend the manuscript for publication

Reviewer 3 Report

The manuscript is improved but I still disagree with the authors on the acceptability of the minimal sample size and while the read depth does provide assurance of accurate transcript identification and quantitation is does not change the fact that comparing two animals to two animals is not sufficient to adequate represent changes in gene expression and as such the results and conclusions are speculative.

The english was improved.

Author Response

The manuscript is improved but I still disagree with the authors on the acceptability of the minimal sample size and while the read depth does provide assurance of accurate transcript identification and quantitation is does not change the fact that comparing two animals to two animals is not sufficient to adequate represent changes in gene expression and as such the results and conclusions are speculative.

Response: Thank you for the review's comments,your comments plays vital role in improving this manuscript. The description of the samples and methods may not accurate, we have corrected in the revised manuscript. In this study, we conduct integrative analysis of transcriptomic and lipidomic profiles. We have referred to some article, such as, Shuji Ueda et.al (2021)., Zhang Jiasu et.al (2021)., and due to research funding limitations,we finally selected 12 samples of subcutaneous adipose from 4 Ningxiang pigs, 4 Berkshire pigs and 4 F1 pigs, respectively, to detect transcriptomic profile. For lipidomic profile, in order to make this research more valuable,we still added some samples even with limited funding after referring to relevant articles. Finally, we selected 18 samples of subcutaneous adipose from 6 Ningxiang pigs, 6 Berkshire pigs and 6 F1 pigs, respectively, to detect lipidomic profile. We also conduct RT-qPCR validation. The current samples has been stored in the -80 refrigerator for 15 months, and is not suitable for detecting transcriptomic profile. We have try ours best to enhance the reliability of results in this study. Thanks for the reviewer.

References

Shuji U, Mana H, Ken-ichi Y, Minoru Y, yasuhito S. Gene expression analysis provides new insights into the mechanism of intramuscular fat formation in Japanese Black Cattle. Genes. 2021.12(8).1107.

Zhang JS, Xu HY, Fang JC, Yin BZ, Wang BB, Pang Z, Xia GJ. Integrated microRNA-mRNA analysis reveals the roles of microRNAs in the muscle fat metabolism of Yanbian cattle. Animal Genetics. 2021.52(5):598-607.

Round 3

Reviewer 3 Report

The authors have missed my point. The limitations of the sample size needs to be acknowledged and the results are speculative and need to be reported as such.

There are still minor issues but the language has been improved.

Author Response

Thank you for the review's comments your comments plays vital role in improving this manuscript. The description of the samples and methods may not accurate, we have corrected in the revised manuscript. In this study, we conduct integrative analysis of transcriptomic and lipidomic profiles. We have referred to some article, such as, Shuji Ueda et.al (2021)., Zhang Jiasu et.al (2021)., and due to research funding limitations we finally selected 12 samples of subcutaneous adipose from 4 Ningxiang pigs, 4 Berkshire pigs and 4 F1 pigs, respectively, to detect transcriptomic profile. For lipidomic profile, in order to make this research more valuable
we still added some samples even with limited funding after referring to relevant articles. Finally, we selected 18 samples of subcutaneous adipose from 6 Ningxiang pigs, 6 Berkshire pigs and 6 F1 pigs, respectively, to detect lipidomic profile. We also conduct RT-qPCR validation. The current samples has been stored in the -80 refrigerator for 15 months, and is not suitable for detecting
transcriptomic profile. We have try ours best to enhance the reliability of results in this study.
References
Shuji U, Mana H, Ken-ichi Y, Minoru Y, yasuhito S. Gene expression analysis provides new insights into the mechanism of intramuscular fat formation in Japanese Black Cattle. Genes. 2021.12(8).1107.
Zhang JS, Xu HY, Fang JC, Yin BZ, Wang BB, Pang Z, Xia GJ. Integrated microRNA-mRNA analysis reveals the roles of microRNAs in the muscle fat metabolism of Yanbian cattle. Animal Genetics. 2021.52(5):598-607.